# Experimental entanglement swapping through single-photon $\chi^{(2)}$ nonlinearity

Yoshiaki Tsujimoto ®[1] ✉, Kentaro Wakui[1], Tadashi Kishimoto[1], Shigehito Miki[2], Masahiro Yabuno ®[2], Hirotaka Terai ®[1,2], Mikio Fujiwara[1] & Go Kato ®[1]

In photonic quantum information processing, quantum operations using nonlinear photon-photon interactions are vital for implementing two-qubit gates and enabling faithful entanglement swapping. However, due to the weak interaction between single photons, the all-photonic realization of such quantum operations has remained out of reach so far. Herein, we demonstrate an entanglement swapping using sum-frequency generation between single photons in a $\chi^{(2)}$-nonlinear optical waveguide. We show that a high signal-to-noise ratio (SNR), stable sum-frequency-generation-based entanglement heralder with an ultralow-dark-count superconducting single-photon detector can satisfy the unprecedented SNR requirement indispensable for the swapping protocol. Furthermore, the system clock is enhanced by utilizing ultrafast telecom entangled photon-pair sources that operate in the GHz range. Our results confirm a lower bound 0.770(76) for the swapped state's fidelity, surpassing the classical limit of 0.5 successfully. Our findings highlight the strong potential of broadband all-single-photonic nonlinear interactions for further sophistication in long-distance quantum communication and photonic quantum computation.

Nonlinear interactions between independent single photons are key to the advancement of photonic quantum information processing[1,2]. In the past few decades, linear optical elements have been commonly adopted to configure such quantum information processing systems[3–6]. While these elements involve the use of general optical components, they require a large number of single-photon sources and detectors as the system sizes increase. The configuration of such systems may be dramatically simplified once the nonlinear photon-photon interaction is unlocked. For example, a complete Bell-state measurement[7], faithful entanglement swapping[8] and even universal quantum computation[9] can be conducted without the need for complex ancillary systems by exploiting $\chi^{(2)}$ interaction between single photons.

In the entanglement swapping protocol, as outlined in ref. 8, a qualitative distinction arises between sum-frequency generation (SFG)-based Bell-state analyzers (SFG-BSA) and those based on linear optics when entangled photon pairs are generated probabilistically.

Specifically, while the quantum states heralded by linear-optical BSAs become mixed states, the states heralded by SFG-BSAs remain close to maximally entangled states. These high-fidelity entangled states can be directly employed in various quantum protocols without the need for postselection. In pursuit of this goal, several experimental attempts have been undertaken to observe SFG between single photons, including SFG between a heralded single photon and weak coherent light[10], as well as between independent heralded single photons[11]. However, quantum operations have remained challenging mainly due to the dark count noise of the single-photon detector, comparable to the SFG signal. Consequently, although there are several feasibility studies[12,13], no quantum operations using SFG between genuine single photons have been achieved thus far.

In this study, we overcome the signal-to-noise ratio (SNR) problem and demonstrate an original entanglement swapping experiment based on SFG between two independent color-distinct single photons. We develop a high SNR and stable SFG-BSA unit, which performs the

[1]National Institute of Information and Communications Technology (NICT), Koganei, Tokyo, Japan. [2]National Institute of Information and Communications Technology (NICT), Kobe, Hyogo, Japan. ✉e-mail: tsujimoto@nict.go.jp

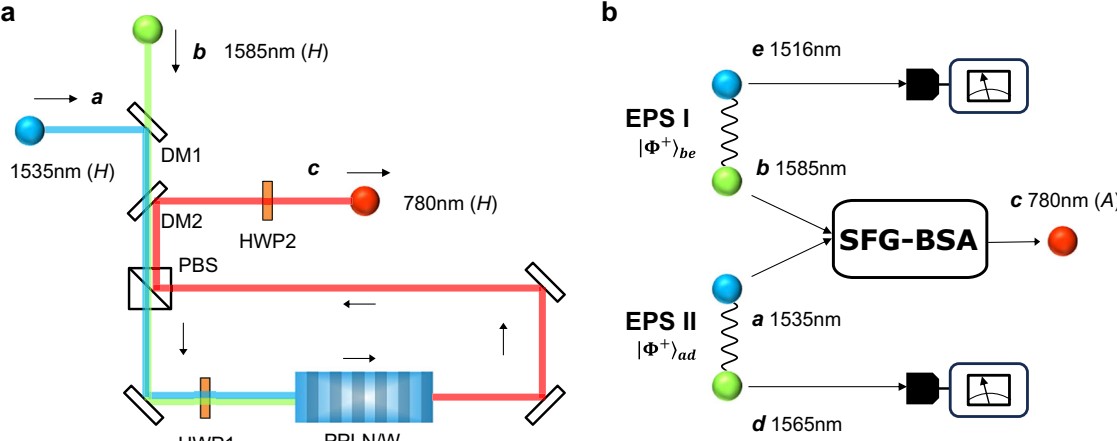

**Fig. 1 | SFG-BSA and SFG-based entanglement swapping. a** Schematic of the SFG-BSA, showing the operation for two *H*-polarized input photons in modes *a* and *b*, which are combined into a single spatial mode by dichroic mirror 1 (DM1). They transmit through a polarization beamsplitter (PBS). Then, they flip to *V* polarization by half-waveplate 1 (HWP1). At the PPLN/W, the *V*-polarized photons in modes *a* and *b* are converted to a *V*-polarized single photon in mode *c* via the SFG process. Finally, the SFG photon is extracted by DM2, and its polarization is flipped back to *H*-polarization by HWP2. **b** Schematic of the SFG-based entanglement swapping. Detection of the *D*-/*A*-polarized SFG photon in mode *c* at 780 nm heralds the creation of entanglement between the photons in modes *d* and *e*.

Bell-state measurement in the polarization degree of freedom with the help of $\chi^{(2)}$ nonlinearity. Our experimental setup features a long periodically-poled lithium niobate waveguide (PPLN/W)[14] within a stable Sagnac interferometer for the SFG-BSA, and an ultralow-dark-count-rate superconducting nanowire single photon detector (SNSPD) optimized for near-infrared detection[15]. Unlike systems that use Mach-Zehnder interferometers (MZIs), an active stabilization is not required in our SFG-BSA unit, enduring weeks-order long-term data acquisition. Moreover, we employ SPDC-based high-repetition-rate telecom entangled photon-pair sources (EPSs)[16] tunable in the GHz range to boost the generation rate of input entangled photon pairs.

First, we performed a quantum-teleportation experiment to assess our SFG-BSA unit. We prepared a weak coherent pulse at the single-photon level and an entangled photon pair as an input state. Then, we confirmed the SFG-BSA high-fidelity operation by teleporting the polarization state of the weak coherent pulse using the detection signal of the SFG photon as a heralder. Subsequently, we conducted the SFG-based entanglement swapping experiment using two entangled photon pairs and the SFG-BSA. To our knowledge, this study is the first to realize a quantum operation using SFG between genuine single photons.

## RESULTS
### The SFG-BSA configuration
Figure 1a shows a schematic of the SFG-BSA setup. A type-0 PPLN/W is incorporated in a Sagnac interferometer to extract two of the four Bell states in the polarization degree of freedom. It should be noted that this degree of freedom was preferred over the time degree of freedom, as the latter incurs photon losses by configuring the measurement system with a passive Franson interferometer[17]. In contrast to MZI-based configurations[18], our SFG-BSA does not require phase stabilization between horizontally (*H*-) and vertically (*V*-) polarized photons. Thus, it can be viewed as a time-reversal of the polarization-entangled photon-pair generation using a Sagnac interferometer with a PPLN/W[16,19,20]. In this configuration (Fig. 1a), no SFG photon is produced if the two input photons are *H*- and *V*-polarized, respectively. However, if both input photons are *V*-(*H*-)polarized, they propagate clockwise (counterclockwise) through the interferometer to produce *V*-(*H*-) polarized SFG photons, respectively.

This can be further discussed through the interaction Hamiltonian:

$$\hat{H} = i\hbar\chi\left(\hat{a}_H\hat{b}_H\hat{c}_H^\dagger + \hat{a}_V\hat{b}_V\hat{c}_V^\dagger\right) + \text{H.c.}, \quad (1)$$

where $\hat{a}_k$ and $\hat{b}_k$ are annihilation operators of single photons with polarization $k \in \{H, V\}$ in the input modes *a* and *b*, respectively, $\hat{c}_k$ is an annihilation operator of a *k*-polarized single photon in the output mode *c*, H.c. is the Hermitian conjugate, and $\hbar$ is the Dirac constant. Here, $\chi \in \mathbb{R}$ is a coupling constant that includes the nonlinear susceptibility of PPLN/W, and it is assumed to be independent of the polarization of the input photons. In addition, we assume that the sum of the photon numbers in the input modes *a* and *b* does not exceed two, and we consider only the lowest order of $\chi$. These assumptions are made to describe the ideal operation of the SFG-BSA. The more realistic situations, including polarization dependency of $\chi$ and multiphoton inputs, are considered in the Theoretical analysis in the Methods section. In the case where one photon is generated in mode *c* as a success event, the corresponding Kraus operator is given by

$$\hat{K} = \sqrt{\eta_{\text{SFG}}}(|H\rangle_c\langle HH|_{ab} + |V\rangle_c\langle VV|_{ab}), \quad (2)$$

where $\eta_{\text{SFG}} := (\chi\tau)^2$ and $|k\rangle_a := \hat{a}_k^\dagger|\text{vac}\rangle$ for $k \in \{H, V\}$ and so on. Here, $\tau$ is the travel time of the input photons through the nonlinear medium and $|\text{vac}\rangle$ is a vacuum state. By detecting a diagonally (*D*-) or anti-diagonally (*A*-) polarized SFG photon in mode *c* (defined by $|D\rangle_c := (|H\rangle_c + |V\rangle_c)/\sqrt{2}$ and $|A\rangle_c := (|H\rangle_c - |V\rangle_c)/\sqrt{2}$), the projection on a Bell state $|\Phi^+\rangle_{ab} := \frac{1}{\sqrt{2}}(|HH\rangle_{ab} + |VV\rangle_{ab})$ or $|\Phi^-\rangle_{ab} := \frac{1}{\sqrt{2}}(|HH\rangle_{ab} - |VV\rangle_{ab})$ will be respectively performed, which corresponds to a quantum parity check.

Compared to a linear-optical BSA[4,5,20–23], the SFG-BSA offers the following unique benefits. First, it aids faithful entanglement swapping based on probabilistic photon pair sources such as SPDC[8], in which detection of an SFG photon deterministically heralds the creation of an entangled photon pair. This makes it well-suited for a long-distance loophole-free Bell test and device-independent quantum key distribution[24]. Interestingly, this principle allows tolerance against optical loss for entanglement swapping (details are given in the next section). Second, by adding another SFG-BSA for odd-parity inputs, it enables a complete Bell-state measurement that distinguishes all four Bell states without employing ancillary systems[7]. Finally, it allows entangling operations between different-color photons, which is useful for building multi-user entanglement networks[13].

Figure 1b describes the entanglement swapping experiment based on the SFG-BSA. EPS I and EPS II generate two maximally entangled photon pairs $|\Phi^+\rangle_{ad} \otimes |\Phi^+\rangle_{be}$ and the SFG-BSA is

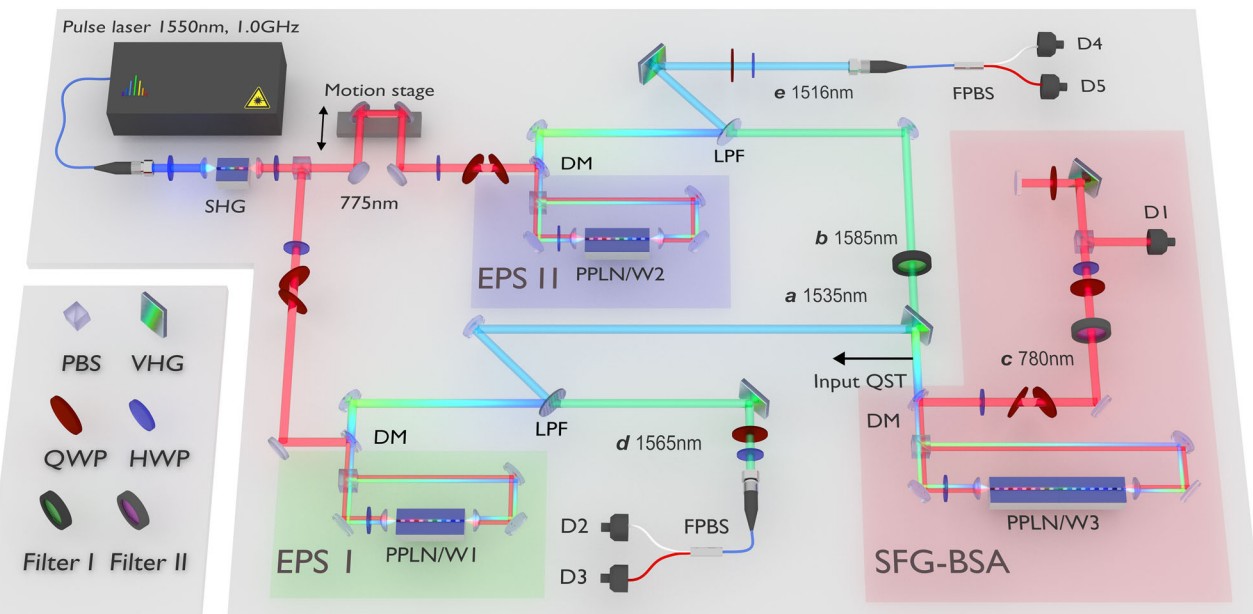

**Fig. 2 | Experimental setup for the SFG-based entanglement swapping.** Pump pulses centered at 775 nm with a 1.0 GHz repetition rate is prepared by second harmonic generation (SHG) of the electro-optic comb centered at 1550 nm and used to pump EPS I and II. Each EPS consists of a PPLN/W in a Sagnac interferometer with a polarizing beamsplitter (PBS). The signal and idler photons at telecom wavelengths are divided into different spatial modes according to low-pass filters (LPFs). The photon in mode $a$ at 1535 nm and the photon in mode $b$ at 1585 nm are narrowed by a volume holographic grating (VHG) and band-pass filter (Filter I), respectively, and they are fed into the SFG-BSA unit. A flip mirror (not shown) just before the SFG-BSA is used to perform the quantum state tomography (QST) on the input quantum states. The output SFG photon in mode $c$ at 780 nm is extracted by a dichroic mirror (DM) and passes through the band pass filter (Filter II) and is diffracted twice by a VHG. The polarization of the SFG photon is projected on the $A$-polarization by means of a quarter-waveplate (QWP), a half-waveplate (HWP), and PBS. The photon-detection signal from the SNSPD (D1) is used as the start signal of a time-to-digital converter (not shown). The photons in modes $d$ and $e$ are diffracted by VHGs. The polarization correlation of the swapped state $\hat{\rho}_{de}$ is evaluated by QWPs, HWPs, and fiber-based PBSs (FPBSs), followed by SNSPDs (D2-D5).

performed on the photons in modes $a$ and $b$, yielding $_c\langle D(A)|\hat{K}|\Phi^+\rangle_{ad} \otimes |\Phi^+\rangle_{be} \propto |\Phi^{+(-)}\rangle_{de}$, respectively. The overall success probability is $\sum_{l\in\{D,A\}} |_c\langle l|\hat{K}|\Phi^+\rangle_{ad} \otimes |\Phi^+\rangle_{be}|^2 = \eta_{\mathrm{SFG}}/2$. The experimental success probability is $\eta_{\mathrm{SFG}}/4$, considering that only one of the $D$- or $A$-polarized portions is measured.

### Robustness of SFG-BSA against optical losses

To see the loss tolerance of the SFG-based entanglement swapping with probabilistic sources, we consider error events of the lowest order, where two photon pairs are generated by EPS I and one photon pair is generated by EPS II. In the presence of high channel loss, the dominant error event is the loss of a single photon among the three photons transmitted through the channel. This type of event can be categorized into two distinct types. First, one photon from EPS I is lost, and one photon from each of EPS I and EPS II successfully arrives at the BSA, which results in a coincidence detection. Second, one photon from EPS II is lost, and a coincidence detection is induced by two photons from EPS I. Although the first error event commonly occurs in the linear-optical BSA and SFG-BSA, the second error event can be excluded using the SFG-BSA. The reason is that the SFG occurs only when at least one photon arrives from each EPS, which results in an increase in the robustness of the SFG-BSA against optical loss. Let $\gamma^2$ be the respective photon pair generation probability of EPS I and EPS II, and let $t$ be the transmittance of each channel. Then, the probabilities of the first and second error events are given by $2\gamma^6 t^2(1-t)$ and $\gamma^6 t^2(1-t)$, respectively. Thus, 1/3 of the error events are excluded using SFG-BSA. More detailed analysis is given in Supplementary Note 1 and 2.

### SFG-BSA unit

The SFG-BSA unit consists of an in-house type-0 MgO-doped PPLN ridge waveguide (PPLN/W3) with free space coupling (length:

**Table 1 | Parameters of the SFG-BSA unit**

|   | $C_{\mathrm{SFG}}$ | $\eta_t$ | $\eta_d$ | $P_a$ | $P_b$ |
|---|---|---|---|---|---|
| $H$ | 2.54 MHz | 0.43 | 0.85 | 80 nW | 61 nW |
| $V$ | 1.94 MHz | 0.40 | 0.85 | 70 nW | 56 nW |

6.3 cm). Details of device fabrication are given in Supplementary Note 5. To estimate the single-photon SFG efficiency, we used coherent light pulses centered at 1535 and 1585 nm. These pulses were generated according to the difference-frequency generation (DFG) at EPS I and II in Fig. 2 with the pulsed pump at 775 nm and the two additional continuous wave lasers at 1565 and 1516 nm, respectively. Their coupling efficiencies to PPLN/W3 were measured to be 0.77 and 0.89, respectively, indicating a large mode matching between the input photon and waveguide modes. The estimated spatial profiles of photons in modes $a$, $b$ and $c$ are shown in Supplementary Note 6. By measuring the average power of the DFG pulses and the count rate of the SFG photons, the internal SFG conversion efficiency for single-photon inputs can be estimated as

$$\eta_{\mathrm{SFG}} = \frac{C_{\mathrm{SFG}}}{\eta_t \eta_d f} \times \frac{hcf}{P_a \lambda_a} \times \frac{hcf}{P_b \lambda_b}, \tag{3}$$

where $c$ is the speed of light, $h$ is Planck's constant, $\eta_t$ and $\eta_d$ are the transmittance of the system and the quantum efficiency of the SNSPD used to detect the SFG photons, respectively. In addition, $P_{a(b)}$ and $\lambda_{a(b)}$ are the average power per mode and wavelength of the DFG pulses in modes $a$ and $b$, respectively. $C_{\mathrm{SFG}}$ is the count rate of the SFG photons, and $f = 1.0$ GHz is the repetition rate of the DFG pulses. Using the experimental parameters summarized in Table 1, we obtained

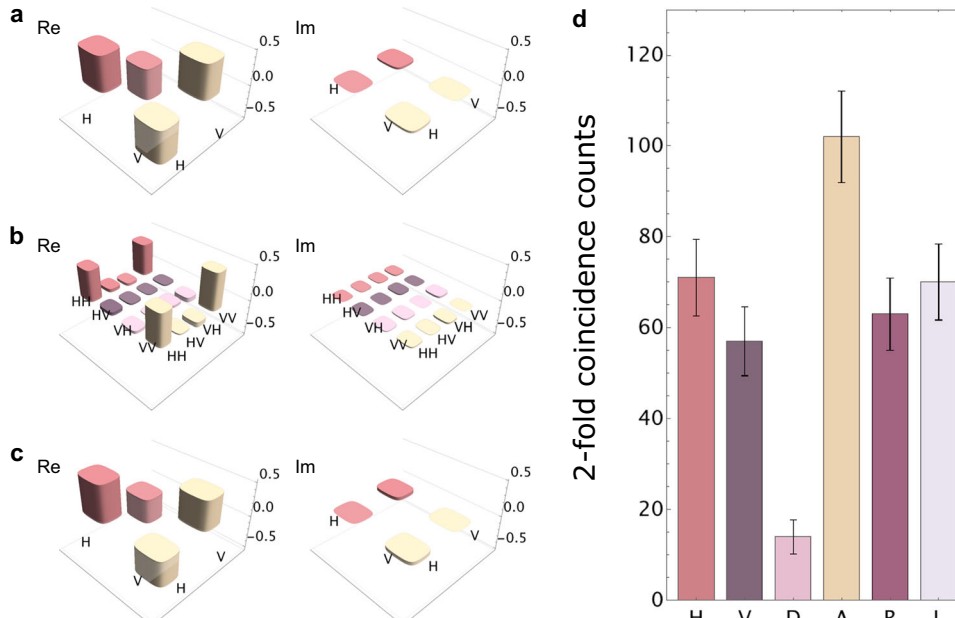

**Fig. 3 | Experimental results for the SFG-based quantum teleportation.**
**a** Density matrix of the *A*-polarized input state. The left (right) of each plot shows the real (imaginary) part of the density matrix, respectively. **b** Density matrix of the entangled state. **c** Density matrix of the teleported state. **d** Raw counts of the teleported photons for the *A*-polarized input light. The measurement time was 13 h for each basis state. Here, *R* and *L* represent right and left circular polarizations, respectively. The error bars were calculated assuming the Poisson statistics.

$\eta_{\mathrm{SFG}}^H = 2.31 \times 10^{-8}$ and $\eta_{\mathrm{SFG}}^V = 2.35 \times 10^{-8}$ for *H*- and *V*-polarized inputs, respectively.

## SFG-based quantum teleportation

To assess our SFG-BSA unit, we performed a quantum teleportation experiment. The input light to be teleported was generated by performing DFG at EPS II. We prepared *H*-, *A*-, and right circularly (*R*-) polarized weak coherent light as input states, and we set the average photon number at 0.95 per pulse. Then, we reconstructed the density matrices of the input states by conducting quantum state tomography[25]. Figure 3a shows an example of the density matrix for the *A*-polarized state (the density matrices of the *H*- and *R*-polarized input states are shown in Supplementary Note 3). The fidelities of the *H*-, *A*- and *R*-polarized states are $\langle H|\hat{\rho}_b|H\rangle = 0.97050(9)$, $\langle A|\hat{\rho}_b|A\rangle = 0.99735(3)$ and $\langle R|\hat{\rho}_b|R\rangle = 0.97500(8)$, respectively, and here $|R\rangle := (|H\rangle + i|V\rangle)/\sqrt{2}$. The fidelity errors correspond to the standard deviations with the assumption of Poisson statistics for the photon counts. In addition, we prepared an entangled photon pair using EPS I, whose density matrix is shown in Fig. 3b. The fidelity to the ideal state $|\Phi^+\rangle$ is $\langle \Phi^+|\hat{\rho}_{ad}|\Phi^+\rangle = 0.9163(4)$. We input the weak coherent light in mode *b* and the photon in mode *a* into the SFG-BSA. Under the condition that a *D*-polarized SFG photon was detected by D1, we performed the quantum state tomography on the photon in mode *d* (see Fig. 2). Figure 3c and d show the density matrix of the teleported state and the raw detection counts for the *A*-polarized input, respectively. The fidelity is 0.890(30), and the measurement time is 13 h for each basis state. For *H*- and *R*-polarized inputs, the fidelities are 0.893(28) and 0.840(39), respectively. These values significantly exceed the classical limit of 2/3[26], indicating the high fidelity of the SFG-BSA for inputs at the single photon level. It is worth noting that the results in ref. 7 were reported as quantum teleportation using SFG with $10^{10}$ input photons on average. Nevertheless, these results can be rather understood as a quantum frequency conversion[27–29] in the weak pump regime, in contrast to the present quantum-teleportation experiment with single-photon-level inputs. This is because in the situation of ref. 7, the input light is regarded as the pump light for the quantum frequency conversion. Hence, the quantum-teleportation-like

behavior is observed under the condition that the conversion efficiency of the quantum frequency conversion can be approximately regarded as proportional to the average intensity of the input light. In the strong-pump regime, where the above approximation does not hold, the polarization state is no longer transferred. A detailed discussion is given in Supplementary Note 4.

## SFG-based entanglement swapping

Figure 2 depicts the experimental setup for entanglement swapping. We prepared two initial entangled photon pairs $\hat{\rho}_{ad}$ and $\hat{\rho}_{be}$ as input states. Their density matrices are shown in Figs. 4a and b, respectively. The fidelities are $\langle \Phi^+|\hat{\rho}_{ad}|\Phi^+\rangle = 0.9079(4)$ and $\langle \Phi^+|\hat{\rho}_{be}|\Phi^+\rangle = 0.8775(6)$. The photons in modes *a* and *b* were combined by a volume holographic grating and fed to the SFG-BSA. When an *A*-polarized SFG photon at 780 nm is detected by the SNSPD (D1), both output ports of the fiber-based polarizing beamsplitter were simultaneously measured using SNSPDs D2, D3, D4, and D5. The low-noise detection of the SFG photon in mode *c* was enabled by D1, which possesses a high quantum efficiency ($\eta_d = 85\%$) and an ultralow dark count rate ($R_d = 0.15$ Hz). The typical quantum efficiency of D2, D3, D4, and D5 was 75%. Figure 4c shows an example of the two-fold coincidence between D1 and D2. We observe a signal peak corresponding to the coincidence detection between photons in modes *c* and *d*. Background counts are primarily caused by the coincidence between the dark count at D1 and the detection of the photon in mode *d* at D2. We used a coincidence window of $\tau_w = 448$ ps, corresponding to seven bins around the signal peak, shown by the red bars in Fig. 4c. The three-fold coincidence histogram among D1, D2, and D4 with bin widths $\tau_w$ is shown in Fig. 4d. We clearly identify a signal peak corresponding to $\langle VV|\hat{\rho}_{de}|VV\rangle$, with a measured SNR equal to $5.57 \pm 1.48$. We performed Z-basis measurements that distinguish the *H*/*V* polarization and X-basis measurements that distinguish the *D*/*A* polarization on the photons in modes *d* and *e*. Each measurement (30 min duration) was repeated 452 times, resulting in an overall measurement time of 226 h for each basis state. The coincidence counts of the swapped state for eight basis states are shown in Figs. 4e and f. The polarization correlation

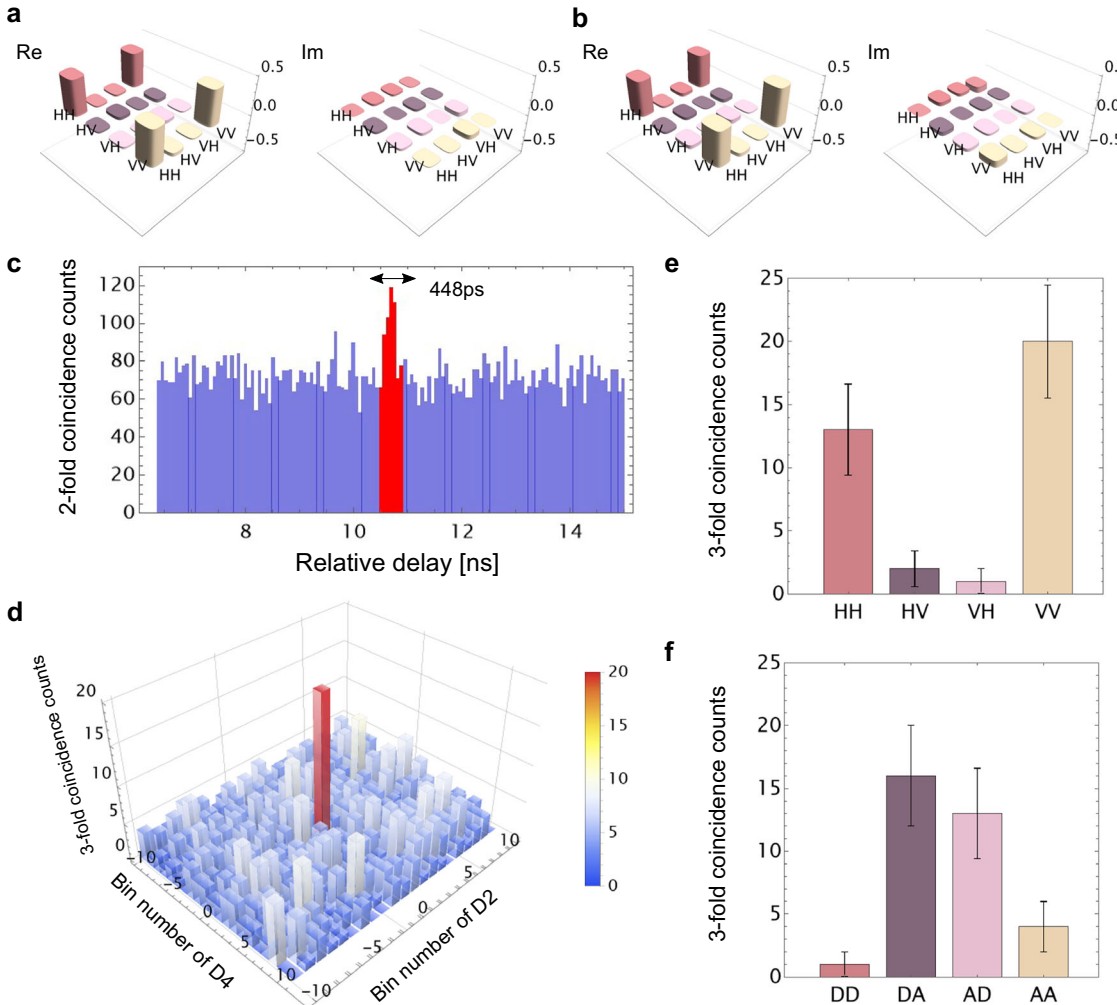

**Fig. 4 | Experimental results for the SFG-based entanglement swapping.**
**a**, **b** Density matrices of the input states from EPS I and II, respectively. **c** Two-fold coincidence counts between D1 and D2. We employed a 448-ps coincidence window corresponding to seven bins around the signal peak. **d** Three-fold coincidence counts among D1, D2, and D4. The center bin corresponds to the signal event. The waveplates were set so that D2 and D4 detect *V*-polarized portions of photons *d* and *e*, respectively. The measurement time was 226 h. **e**, **f** Raw detection counts of the swapped state for each basis state. The error bars were calculated assuming the Poisson statistics.

corresponding to the target swapped state $|\Phi^-\rangle_{de} := (|HH\rangle_{de} - |VV\rangle_{de})/\sqrt{2} = (|DA\rangle_{de} + |AD\rangle_{de})/\sqrt{2}$ is clearly identified. We evaluated the quality of the swapped state by the visibilities for Z-basis ($V_Z := \langle \hat{Z}_d \hat{Z}_e \rangle$) and X-basis ($V_X := -\langle \hat{X}_d \hat{X}_e \rangle$), where $\hat{Z} := |H\rangle\langle H| - |V\rangle\langle V|$ and $\hat{X} := |H\rangle\langle V| + |V\rangle\langle H|$ are Pauli operators. We observed high visibilities of $V_Z = 0.833(92)$ and $V_X = 0.706(121)$. Although these parameters are not sufficient to reconstruct the complete density matrix, they help to estimate the lower bound of the fidelity $F \geq F_{low} = (V_Z + V_X)/2$[30,31]. The experimental value of $F_{low}$ was found to be equal to 0.770(76), which confirms that the swapped state is strongly entangled.

## DISCUSSION

We compare the performance of the presented approach based on the SFG-BSA with existing linear optical approaches. As outlined in the theoretical proposal[8], a key application of the SFG-BSA is the heralded generation of entangled photon pairs using probabilistic photon sources, a method wherein the preparation of an entangled photon pair is conditioned on the detection of auxiliary photons. Specifically, upon the detection of a heralding signal, the entangled photon pair is deterministically prepared and can subsequently be utilized in various quantum protocols. The linear optics-based approach[32] involves the preparation of three pairs of twin photons (six photons in total), utilizing four of these photons to herald the generation of an entangled photon pair. As discussed in previous studies[8,32], there exists a trade-off between the success probability and the fidelity of the heralded state. In this context, we compute the maximum success probability that ensures a fidelity greater than 0.9 using the equations provided in ref. 8. Assuming a photon detection efficiency of 0.7, the success probability for the linear optical implementation is calculated to be $1.5 \times 10^{-11}$. In comparison, using the SFG-BSA, our theoretical model yields a success probability of $1.2 \times 10^{-11}$ with an SFG efficiency of $2.3 \times 10^{-8}$ and a photon detection efficiency of 0.7, which is nearly identical to the linear optical approach. Here, we assume that a complete Bell-state measurement is performed at the SFG-BSA. Furthermore, in regions with lower detection efficiency, the SFG-BSA becomes more efficient. For instance, at a photon detection efficiency of 0.5, the SFG-BSA exhibits a performance 16 times more efficient than the linear optics-based approach.

Another important application of the SFG-BSA is its use in the heralding scheme for a loophole-free Bell test with probabilistic photon sources. Upon detection of the heralding signal, the detection loophole induced by transmission losses can be effectively closed. In this context, a linear optical BSA has been identified as one of the plausible schemes for achieving such a heralded Bell test[33,34]. However,

**Table 2 | Parameters of ESP I and EPS II**

| | EPS I | | | EPS II | | |
|---|---|---|---|---|---|---|
| | $\mu_1$ | $\eta_1$ | $t_1$ | $\mu_2$ | $\eta_2$ | $t_2$ |
| H | 0.060 | 0.097 | 0.44 | 0.080 | 0.070 | 0.56 |
| V | 0.050 | 0.11 | 0.48 | 0.061 | 0.10 | 0.57 |

a qualitative difference exists between the quantum states heralded by the linear optical BSA and those heralded by the SFG-BSA. In a Bell test experiment, the correlation of the measurement outcomes is typically characterized by the parameter $S$, with the Bell-CHSH inequality stipulating that $|S| < 2$ within the framework of local realism. According to ref. 34, using the linear optical BSA as a heralding scheme, the maximum achievable value of $S$ was shown to be 2.34. Furthermore, it was shown that a detection efficiency of at least 0.911 is required for both Alice's and Bob's detectors to observe a violation of the Bell-CHSH inequality. In contrast, our simulation results indicate that, when employing the SFG-BSA approach, the achievable value of $S$ approaches the Tsirelson bound of $2\sqrt{2}$[35]. In addition, we find that a detection efficiency of 0.68 is sufficient to observe the Bell-CHSH inequality violation, significantly reducing the stringent requirements for Alice and Bob's detection systems compared to the linear optical BSA approach. These results can be attributed to the fact that the fidelity of the quantum state heralded by the SFG-BSA can approach unity, whereas that of the state heralded by the linear optical BSA is fundamentally limited to a maximum of 1/2 in the absence of postselection[8].

Finally, we examine the extent to which the efficiency of SFG must be improved for future applications. The SNR at the SFG photon detection achieved by our current system is insufficient to herald an entangled state capable of violating the Bell-CHSH inequality. However, theoretical simulations using experimental parameters from Tables 1 and 2 indicate that tripling the SFG efficiency would result in an adequate SNR to herald the quantum state that exhibits a violation of the Bell-CHSH inequality. For the application to DIQKD, a further enhancement of the SFG efficiency by a factor of 50 would be required. This level of improvement is considered possible, given recent advancements in nonlinear resonators, which have demonstrated increases in efficiency by two to three orders of magnitude compared to PPLN/W[36,37]. (For further details on these simulations, see Supplementary Note 7.)

In conclusion, we have implemented a BSA using SFG between single photons and demonstrated the quantum teleportation and entanglement swapping. We have confirmed the high fidelity of the teleported/swapped states, which is an important milestone toward quantum information processing using single-photon $\chi^{(2)}$ nonlinearity. Toward practical applications, SFG efficiency improvements of several orders of magnitude are needed. Nevertheless, our simulation shows that even a several-fold improvement in the SFG efficiency is useful for the loophole-free Bell test. Recently, there has been notable progress in the research on nonlinear devices. Combined with these technologies, our method provides a new avenue toward all-photonic quantum information processing using the single-photon $\chi^{(2)}$ nonlinearity.

## Note added
During the preparation of the manuscript presented here, we learned of an experiment demonstrating quantum teleportation using single-photon SFG by Akin et al.[38].

## METHODS
### Entangled photon pair sources
We employed an electro-optic frequency comb generator to emit a fundamental pulse to pump SPDC and realized a high-speed generation of entangled photon pairs. The setups of EPS I and EPS II are shown

on the left side of Fig. 2. We generated a frequency comb centered at 1550 nm with a 10 GHz repetition rate by electro-optic phase modulation on the narrow-band continuous wave laser light[16,39–41]. We reduced it to 1/10 through optical gating[42]. We selected a 1.0 GHz repetition rate such that the SNSPDs can resolve each pulse without saturating. Pump pulses at 775 nm for SPDC were generated by SHG using a 1.0 cm-long PPLN/W. The SHG spectral bandwidth was measured to be 34 GHz full-width at half maximum (FWHM), and the power was stabilized during the experiment. Each EPS consists of a 3.4 cm-long PPLN/W with a Sagnac configuration[16,20]. The pump power coupled to the PPLN/W was approximately 1.0 mW for both the clockwise and counterclockwise directions. We used volume holographic gratings with a 55-GHz FWHM (i.e., in accordance with the phase-matching bandwidth of the SFG-BSA unit) to narrow the bandwidths of the signal and idler photons. The spectral distributions of the photons in modes $a$, $b$ and $c$ are shown in Supplementary Note 6. Table 2 summarizes the average photon numbers per mode ($\mu$), Klyshko efficiencies[43] ($\eta$), and the transmittance ($t$) in the optical circuit before the SFG-BSA, including the coupling efficiencies to the SFG-BSA. We employed a flip mirror (not shown) just before the SFG-BSA to evaluate the input quantum states from EPS I and II. The detection rate of the entangled photon pairs for each EPS was measured to be 1.1 MHz.

### Theoretical SFG efficiency
PPLN/W3 used for SFG possesses a crystal length of $L = 6.3$ cm and a core diameter of 7.2 µm × 8.0 µm[14]. The normalized SHG efficiency was measured to be $\eta_{\text{SHG}} = 28\% \cdot \text{W}^{-1} \cdot \text{cm}^{-2}$ using a narrow-band continuous-wave laser light. The theoretical SFG conversion efficiency for single-photon inputs can be estimated using the following equation[8,10]:

$$\eta_{\text{SFG}}^{\text{th}} = \frac{\eta_{\text{SHG}}}{2} \times \frac{hc}{\lambda} \times \frac{\Delta\hat{\nu}L}{\text{tbp}}, \tag{4}$$

where $\Delta\hat{\nu}$ is the spectral acceptance of the crystal and tbp is the time-bandwidth product of the SFG photon. By substituting the experimental values of $\lambda = 1560$ nm, $\Delta\hat{\nu} = 2.48 \times 10^2$ GHz · cm and tbp = 0.67, we obtain $\eta_{\text{SFG}}^{\text{th}} = 4.16 \times 10^{-8}$. The SFG efficiency for SPDC photons is estimated considering the overlap integral among the spectral distributions of the input photons $\phi_{a(b)}(\lambda_{a(b)})$, and the normalized phase-matching function of PPLN/W3 $\eta(\lambda_a, \lambda_b)$ as[10]

$$\eta_{\text{SFG}}^{\text{eff}} = \eta_{\text{SFG}}^{\text{th}} \int \int \phi_a(\lambda_a)\phi_b(\lambda_b)\eta(\lambda_a, \lambda_b)d\lambda_a d\lambda_b. \tag{5}$$

From the measured results, when $\phi_a(\lambda_a)$, $\phi_b(\lambda_b)$, $\eta(\lambda_a, \lambda_b)$ are calculated as Gaussians with FWHM of 0.31, 0.33, and 0.080 nm, respectively, we obtain $\eta_{\text{SFG}}^{\text{eff}} = 2.42 \times 10^{-8}$, which is close to the experimental values of $\eta_{\text{SFG}}^H = 2.31 \times 10^{-8}$ and $\eta_{\text{SFG}}^V = 2.35 \times 10^{-8}$.

### Widths of coincidence windows
The width of the coincidence window is determined according to the timing jitters of the SNSPDs. The SFG photon in mode $c$ at 780 nm is detected by D1, whose timing jitter is 190 ps. The telecom photons are detected by D2, D3, D4 and D5, whose timing jitters are 48, 98, 64, and 162 ps, respectively (Fig. 2). The largest timing jitter among the three-fold coincidences is given according to the convolution of the timing jitters of D1, D3, D5 and the coherence time of SPDC photons of 14 ps as 269 ps. We used the coincidence window with $\tau_w = 448$ ps, which covers 96% of the signal events.

### Theoretical analysis of visibilities.
We discuss the validity of the experimentally obtained visibilities, using a realistic model and independently measured experimental parameters. We consider the dark counts in D1 and the multiple photons generated in EPS I and EPS II using the following approach. First, we consider the events where a

maximum of three photon pairs is produced between EPS I and EPS II as an input. Then, we perform the SFG operation after the photons in modes $a$ and $b$ experienced system losses. We assume that the coupling constant $\chi$ in Eq. (1) is dependent on the inputs' polarization, which is defined as $\chi_{H(V)}$ for $H$-($V$-) polarized input, respectively. In this model, a maximum of one SFG photon is produced, which is detected by a threshold detector D1. Conditioned by the detection signal from D1, polarization correlation measurements were performed using D2, D3, D4 and D5. The three-fold coincidence probability is given by $P_{ij}^{SFG}(\theta_1, \theta_2)$ with $i, j \in \{H, V\}$. Here, Z-basis and X-basis are expressed by $(\theta_1, \theta_2) = (0, 0)$ and $(\pi/4, \pi/4)$, respectively. For example, $P_{HV}^{SFG}(\pi/4, \pi/4)$ corresponds to the coincidence probability where $D$- and $A$-polarized photons are detected by D3 and D4 according to the heralding signal from D1, respectively. We also consider the event where the heralding signal from D1 is caused by the dark counts. Because the dark count probability at D1 is given by $R_d \times \tau_W$, the coincidence detection probability between the photons in modes $d$ and $e$ heralded by the dark count at D1 is given by $R_d \tau_W (P_{ij}^{Acd}(\theta_1, \theta_2) - P_{ij}^{SFG}(\theta_1, \theta_2))$, where $P_{ij}^{Acd}(\theta_1, \theta_2)$ is the accidental coincidence detection probability between the photons in modes $d$ and $e$. The visibilities of the swapped state are calculated using the above coincidence probabilities as

$$V_Z^{th} = \frac{P_{HH}(0,0) + P_{VV}(0,0) - P_{HV}(0,0) - P_{VH}(0,0)}{P_{HH}(0,0) + P_{VV}(0,0) + P_{HV}(0,0) + P_{VH}(0,0)} \quad (6)$$

and

$$V_X^{th} = \frac{P_{HV}(\frac{\pi}{4}, \frac{\pi}{4}) + P_{VH}(\frac{\pi}{4}, \frac{\pi}{4}) - P_{HH}(\frac{\pi}{4}, \frac{\pi}{4}) - P_{VV}(\frac{\pi}{4}, \frac{\pi}{4})}{P_{HV}(\frac{\pi}{4}, \frac{\pi}{4}) + P_{VH}(\frac{\pi}{4}, \frac{\pi}{4}) + P_{HH}(\frac{\pi}{4}, \frac{\pi}{4}) + P_{VV}(\frac{\pi}{4}, \frac{\pi}{4})}, \quad (7)$$

where $P_{ij}(\theta_1, \theta_2) = P_{ij}^{SFG}(\theta_1, \theta_2)(1 - R_d \tau_W) + R_d \tau_W P_{ij}^{Acd}(\theta_1, \theta_2)$. The detailed calculation and the theoretical model are presented in Supplementary Note 1.

By substituting $\eta_{SFG}^H = 2.31 \times 10^{-8}$, $\eta_{SFG}^V = 2.35 \times 10^{-8}$, $R_d = 0.15$, $\tau_W = 448 \times 10^{-12}$ and the parameters in Tables 1 and 2, the theoretically calculated visibility values are $V_Z^{th} = 0.78$ and $V_X^{th} = 0.76$, respectively, which exhibit a good agreement with the experimentally observed visibilities $V_Z = 0.833(92)$ and $V_X = 0.706(121)$.

## Data availability

The authors declare that the data supporting the findings of this study are available within the paper, its supplementary information files, and Figshare [https://doi.org/10.6084/m9.figshare.29919704].

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

## Acknowledgements

Y.T. thanks Rikizo Ikuta and Toshiki Kobayashi for helpful discussions. This work was supported by the Japan Society for the Promotion of Science (JP18K13487, JP20K14393, JP22K03490) and R&D of ICT Priority Technology Project (JPMI00316).

## Author contributions

Y.T. conceived the idea, conducted the experiment with the assistance of K.W., and performed the simulation with the assistance of G.K.. K.W., G.K., and M.F. provide valuable advice and discussions on the theory and experiment. T.K. fabricated PPLN/W3 used for the SFG-BSA unit. S.M., M.Y., and H.T. developed the SNSPD system. Y.T. wrote the manuscript with the help of all the other authors.

## Competing interests

The authors declare no competing interests.
