## [Transparent Peer Review file · Nature Communications]

Experimental entanglement swapping through single-photon $\chi^{(2)}$ nonlinearity

Corresponding Author: Dr Yoshiaki Tsujimoto

A version of this paper was originally rejected for publication by Nature Communications, however that decision was reconsidered after appeal by the authors.

Version 1:

Reviewer comments:

Reviewer #1

(Remarks to the Author)

The authors experimentally demonstrate the ability to distinguish between the two “bunched” Bell-states of photonic polarisation qubits using a Sagnac set-up with a $\chi^{(2)}$ non-linearity. Using only linear optics (and no ancilla photons) it is not possible to simultaneously distinguish between the two “bunched” and two “anti-bunched” Bell-states. Although they only demonstrate the hard, “bunched” case they argue that a straightforward extension of their set-up would allow all four Bell-states to be distinguished. To demonstrate that their device is working at the quantum level they use it to teleport and entanglement swap photonic polarisation qubits with fidelities exceeding the classical bound.

The paper is clearly written. The theoretical design and analysis appears correct and the experimental data appears to have been obtained in a rigorous manner and analysed correctly, taking into accounts errors. The claimed scientific results seem well justified.

The major technical advances in this paper seem to be the increased stability of the non-linear optical analyser set-up and the very low dark-count rates of the single photon detectors. It does not appear that the non-linearity of the source is significantly increased over previous work and so the efficiency of the Bell-state analyser is very low. The breakthrough in getting the results is having the stability good enough and the dark counts low enough that they can operate at the single photon quantum level. However, in order for the device to be useful for most applications, e.g. quantum computing gates or quantum repeaters, the efficiency would need to be many orders of magnitude higher. Because advances in the efficiency have not been made the importance of the results is not so clear to me. In the conclusion it is stated that “our method” could be used for loop-hole free Bell tests and DI-QKD – do you mean using the current set-up, in spite of the very low efficiency? Perhaps with some minor modifications? Or do you mean a future set-up where efficiencies are much higher? Some more explanation of this point would be useful.

Reviewer #2

(Remarks to the Author)

The manuscript of Yoshiaki Tsujimoto et al. titled "Experimental entanglement swapping through single-photon $\chi^{(2)}$ nonlinearity" presents the breakthrough experimental demonstration of quantum teleportation and entanglement swapping using the parametric nonlinear process of sum-frequency generation (SFG).

The SFG process has previously been successfully performed for single-photon states. Still, the full demonstration of entanglement swapping using SFG or single-photon nonlinear optical interaction, for that matter, has not been implemented so far. The presented experiment sets a new frontier of applying photon-photon nonlinearity in quantum information processing.

The success probability of entanglement swapping is extremely low ($\sim 10^{-8}$), leading to long integration times (226 hours for each basis state of swapped state measurement). This severely limits the feasibility of using such an approach in a

practical quantum network.

The success rate of the experiment is so low that the Authors cannot perform the complete quantum state characterization of the swapped state. However, they measured the lower bound of the swapping fidelity exceeding the classical limit, i.e. demonstrating the quantum regime.

While the paper highlights advantages over linear-optical Bell-state measurements, it does not compare against atomic/ion-based nonlinear interactions or other frequency conversion schemes. Furthermore, the Authors completely omitted discussion of/comparison to linear-optical schemes employing auxiliary photons and auxiliary degrees of freedom (hyperentanglement) exceeding 50% success probability of Bell-state measurement. We would expect the comparison of overall performance (e.g., overall teleportation and swapping rates) to other established approaches.

The manuscript is well-structured and technically precise. Some explanations in the main text are dense, particularly in describing the experimental setup and theoretical background. The highly technical nature of the work, particularly the nonlinear optics and entanglement swapping protocol, may be challenging for non-specialists. The main text is overloaded with an excessive number of acronyms and abbreviations. The discussion section lacks clarity on how the work compares quantitatively with other approaches and its realistic path toward practical use.

In our opinion, the Authors demonstrated a nice and highly challenging piece of quantum technology, which may draw the attention of the community, even though the performance of the SFG-based Bell-state analyzer, quantum teleportation, and entanglement swapping in real-world quantum communication protocols is yet quite far from being applicable. Unfortunately, the manuscript is very hard to read, especially for readers outside the quantum optics field. Consequently, the paper does not yet meet the broad interest threshold of Nature Communications without significant revisions. If the authors improve clarity, broaden accessibility, and better justify the impact, it might be considered.

Reviewer #3

(Remarks to the Author)

Reviewer #4

(Remarks to the Author)

The manuscript "Experimental entanglement swapping through single-photon $\chi(2)$ nonlinearity" presents the first entanglement swapping experiment based on the nonlinear Bell-state analyser (sum frequency generation) operating at the single-photon level.

Compared to previous experiments, this work involves two individual photons in the SFG process and demonstrates feasibility by directly measuring the polarisation utilising superconducting nanowire single-photon detectors (SNSPDs). In the past, there were two main obstacles hindering the progress in this direction, namely the low efficiency of SFG at the single-photon level and the low efficiency of SNSPDs.

To overcome these challenges, the authors constructed a home-made doped PPLN waveguide and used SNSPD with a high quantum efficiency.

The presented experimental results clearly demonstrate faithful entanglement swapping between photons and are well-supported by theoretical calculations including losses.

Such a demonstration of quantum protocols based on nonlinear single-photon interactions may open the way to all-optical quantum information processing and communication.

I would recommend to publish the paper after authors address the following minor comments:

- The current efficiency of SFG process is about 10^{-8} which is slightly higher but comparable to previous single-photon SFG experiments [Phys. Rev. Lett. 113, 173601, 2014]. Is it possible to improve this efficiency in PPLN waveguides by better design? Or are the new materials and devices such as nanophotonic cavities [<https://arxiv.org/pdf/2411.15437>] the only way to enhance the strength of nonlinear interaction ?

- In the section "Robustness of SFG-BSA against optical losses", two types of fake success events are mentioned, but the probability only for the second one is shown. It would be useful to estimate the first one as well.

- It would be helpful to present the mode profiles of the PPLN waveguides involved in SFG process and add more information about spectral characteristics of the interacting photons.

Version 2:

Reviewer comments:

Reviewer #1

(Remarks to the Author)

The authors have done a good job of answering the questions and comments posed by myself and the other reviewers. They have modified the manuscript so as to make it more complete and place its importance better in context of the current state of the art. Their comments are persuasive that this research does indeed represent a significant advance and on balance I am now happy to recommend publication.

Reviewer #2

(Remarks to the Author)

The authors have addressed all the concerns raised in our previous report. In their revised paper, they add a detailed comparison with existing approaches, include the relevant references, and expand both the Discussion section and the Supplementary Materials. The manuscript is now significantly improved, and we recommend it for publication in Nature Communications in its present form.

A minor issue remains: The main text still uses too many acronyms, which may negatively impact its readability. Examples of acronyms that can be easily omitted: 5 hits - DFG, VHG; 4 hits - QIP; 3 hits - QST, QFC; 2 hits - DIQKD, CW.

Reviewer #3

(Remarks to the Author)

Reviewer #4

(Remarks to the Author)

The Authors have answered all the questions, the manuscript can be published in the current form.

RESPONSES TO REVIEWERS

Reviewer 1

We would like to thank the reviewer for the effort and time spent reading our manuscript and posing questions and comments, which improve on describing future prospects for practical applications. Below, we address the reviewer's comments and questions.

- The authors experimentally demonstrate the ability to distinguish between the two “bunched” Bell-states of photonic polarisation qubits using a Sagnac set-up with a $\chi^{(2)}$ non-linearity. Using only linear optics (and no ancilla photons) it is not possible to simultaneously distinguish between the two “bunched” and two “anti-bunched” Bell-states. Although they only demonstrate the hard, “bunched” case they argue that a straightforward extension of their set-up would allow all four Bell-states to be distinguished. To demonstrate that their device is working at the quantum level they use it to teleport and entanglement swap photonic polarisation qubits with fidelities exceeding the classical bound. The paper is clearly written. The theoretical design and analysis appears correct and the experimental data appears to have been obtained in a rigorous manner and analysed correctly, taking into account errors. The claimed scientific results seem well justified.

We thank the reviewer for highlighting that our scientific results are rigorously justified.

- The major technical advances in this paper seem to be the increased stability of the non-linear optical analyser set-up and the very low dark-count rates of the single photon detectors. It does not appear that the non-linearity of the source is significantly increased over previous work and so the efficiency of the Bell-state analyser is very low. The breakthrough in getting the results is having the stability good enough and the dark counts low enough that they can operate at the single photon quantum level. However, in order for the device to be useful for most applications,

e.g. quantum computing gates or quantum repeaters, the efficiency would need to be many orders of magnitude higher. Because advances in the efficiency have not been made the importance of the results is not so clear to me.

The reviewer pointed out that the nonlinear efficiency achieved in our work is comparable to that of previous studies. If nonlinear efficiency were to be improved, the primary benefit would likely be a reduction in the time required for data acquisition. While this is a valid observation, we would like to emphasize that the anticipated improvement in nonlinear efficiency and the key technological advancements realized in this study—namely, the long-term operational stability of the Bell state analyzer and the low-noise photon detection, both of which were acknowledged by the reviewer—, serve an fundamentally equivalent function in achieving the objective demonstrated in this study. Therefore, it is more important to focus on the fact that **a very new task i. e., quantum operation via single-photon $\chi^{(2)}$ nonlinearity has been successfully demonstrated by crossing a critical SNR threshold that must first be met before any application is possible.** Moreover, the results presented here represent a significant milestone that can serve as a benchmark for future device development.

- In the conclusion it is stated that “our method” could be used for loop-hole free Bell tests and DI-QKD – do you mean using the current set-up, in spite of the very low efficiency? Perhaps with some minor modifications? Or do you mean a future set-up where efficiencies are much higher? Some more explanation of this point would be useful.

To apply our system for the loophole-free Bell tests and DIQKD, a future set-up with higher nonlinear efficiency is necessary. Thanks to the reviewer’s comments, we were able to conduct additional theoretical simulations, which led to the establishment of valuable benchmarks. Specifically, it was revealed that a 50-fold improvement in nonlinear efficiency would enable the application of our system to DIQKD. This improvement was shown to be within the range achievable with state-of-the-art nonlinear resonators.

The main obstacle in loophole-free Bell tests and DIQKD is that the optical loss in the transmission channel creates a detection loophole. To circumvent the impact of transmission losses, the single-photon SFG can be used as a “herald” as proposed in Ref. [1]. To this end, the heralding signal caused by the SFG-photon detection must be at least comparable to the dark count rate of the detector. Under the “fake” heralding signal caused by the dark count, the vacuum state and maximally mixed states are shared between Alice and Bob, which disturbs the violation of the Bell’s inequality. We note that Alice and Bob cannot discard these unwanted events, since loophole-free Bell tests must be performed without such postselections.

Unfortunately, the SNR at the SFG-photon detection achieved by our current system is not enough to observe the violation of Bell’s inequalities without the loopholes. Nevertheless, the theoretical simulation with the experimental

parameters in Tables 1 and 2 of the manuscript reveals that the three times higher SFG efficiency can achieve the sufficient SNR to observe the violation. To apply for DIQKD, a further improvement of SFG efficiency, i.e., by a factor of 50, is needed, which should be a possible level of improvement in light of recent progress in nonlinear resonators reporting efficiency improvements of two to three orders of magnitude compared to PPLN/W [2, 3]. We added these discussions in the Discussion section, and detailed explanation in Supplementary Note 7.

Reviewer 2

We would like to thank the reviewer for the effort and time spent reading our manuscript and posing comments about clarity and comparison with existing approaches, which improve the appealing nature of our manuscript. Below, we address the reviewers' comments.

- The manuscript of Yoshiaki Tsujimoto et al. titled “Experimental entanglement swapping through single-photon $\chi(2)$ nonlinearity” presents the breakthrough experimental demonstration of quantum teleportation and entanglement swapping using the parametric nonlinear process of sum-frequency generation (SFG). The SFG process has previously been successfully performed for single-photon states. Still, the full demonstration of entanglement swapping using SFG or single-photon nonlinear optical interaction, for that matter, has not been implemented so far. The presented experiment sets a new frontier of applying photon-photon nonlinearity in quantum information processing. The success probability of entanglement swapping is extremely low ($\sim 10^{-8}$), leading to long integration times (226 hours for each basis state of swapped state measurement). This severely limits the feasibility of using such an approach in a practical quantum network. The success rate of the experiment is so low that the Authors cannot perform the complete quantum state characterization of the swapped state. However, they measured the lower bound of the swapping fidelity exceeding the classical limit, i.e. demonstrating the quantum regime.

We thank the reviewers for mentioning that “The presented experiment sets a new frontier of applying photon-photon nonlinearity in quantum information processing”.

- While the paper highlights advantages over linear-optical Bell-state measurements, it does not compare against atomic/ion-based nonlinear interactions or other frequency conversion schemes. Furthermore, the Authors completely omitted discussion of/comparison to linear-optical schemes employing auxiliary photons and auxiliary degrees of freedom (hyperentanglement) exceeding 50% success probability of Bell-state measurement. We would expect the comparison of overall performance (e.g., overall teleportation and swapping rates) to other established approaches.

We thank the reviewers for this comment. We strongly agree that comparison with existing approaches will be useful to improve the impact of the manuscript. It is noteworthy that we focused on the all-photon approaches since, as far as we know, no entanglement swapping of photons based on atomic nonlinearities have been reported so far.

As rightly pointed out by the reviewer, several prior studies have reported Bell-state analyzers (BSAs) surpassing the 50% success probability using hyperentanglement [4, 5] and ancillary photons [6, 7, 8, 9]. According to the linear optical scheme proposed in [7], the success probability of the Bell-state measurement is boosted by introducing auxiliary entangled photons. For example, to discriminate 75% (87.5%) of the Bell states, four (twelve) auxiliary photons are necessary, respectively. Therefore, the overall success probability is strongly dependent on the generation and detection probabilities of auxiliary entangled photons. We can then compare the efficiency orders of magnitude in our SFG-BSA and such linear optical BSAs (LO-BSAs) using ancillary photons. First, as demonstrated in [8], when preparing entangled photons via SPDC, the overall success probabilities required to distinguish Bell states at 75% and 87.5% are estimated to be approximately $\eta_{\text{BSA}}^{75\%} = \gamma^4 \eta_d^6$ and $\eta_{\text{BSA}}^{87.5\%} = \gamma^{12} \eta_d^{14}$, respectively [7]. Here, γ^2 and η_d are the SPDC generation probability and photon detection efficiency, respectively. Substituting $\gamma^2 = 0.01\text{--}0.05$, which are typical values set in the experiments, and $\eta_d = 0.7$, we obtain $\eta_{\text{BSA}}^{75\%} \sim 10^{-5}$ to 10^{-4} and $\eta_{\text{BSA}}^{87.5\%} \sim 10^{-15}$ to 10^{-10} . On the other hand, SFG-BSA allows discriminating 100% of Bell states with success probability of $\eta_{\text{SFG}} \eta_d = 1.6 \times 10^{-8}$ with the SFG efficiency of 2.3×10^{-8} used in our experiment and assuming $\eta_d = 0.7$ as before. Thus, our SFG-BSA is already a strong contender for LO-BSA in terms of the success probability. For a more precise numerical comparison, it is necessary to derive the analytical solution for $\eta_{\text{BSA}}^{75\%}$ and $\eta_{\text{BSA}}^{87.5\%}$ in LO-BSA, including contribution of higher order photon pairs using SPDC; however, this is beyond the scope of this paper and should be addressed in future studies.

The preceding discussion has focused on a comparison in terms of performance as a BSA. However, the SFG-BSA possesses another critical aspect—its capability to serve as an entanglement herald. Specifically, the SFG-based BSA was theoretically proposed [1] to herald the **deterministic** generation of entanglement using probabilistic photon pair sources—referred to as heralded generation of entangled photon pairs—**a task that conventional LO-BSAs cannot accomplish**. This unique feature, achieved by removing fake successful events using SFG as shown in Fig. 1, enables the direct use of the entangled pair in subsequent processes without postselection and is particularly advantageous for mitigating transmission losses in long-distance loophole-free Bell tests. The qualitative difference between the LO-BSA and the SFG-BSA arises from the fact that the fidelity of the quantum state heralded by the SFG-BSA can approach unity, whereas that of the state heralded by the LO-BSA is fundamentally limited to a maximum of 1/2 in the absence of postselection [1].

Fortunately, there is a linear optical method to prepare an entangled photon pair deterministically. However, it requires a scheme significantly more

Figure 1: **Entanglement swapping using LO-BSA and SFG-BSA.** **a**, A successful event in entanglement swapping using LO-BSA. **b**, **c**, “Fake” successful events in entanglement swapping using LO-BSA. The final state heralded by the LO-BSA is given by the classical mixture of the three events **a**, **b** and **c** because the LO-BSA cannot distinguish these three events without postselecting coincidence between photons A1 and B2. **d**, Entanglement swapping using SFG. The “fake” successful events such as **b** and **c** are rejected.

complex than the LO-BSA. Such a linear optical scheme was theoretically proposed in [10] and the proof-of-principle experiments have been demonstrated in [11, 12], where three pairs of twin photons (six photons in total) are prepared, and four of these photons are used to herald the generation of an entangled photon pair. According to [1] and [10], there is a trade-off between the fidelity and the success probability. We calculate the maximum success probability to achieve the fidelity larger than 0.9 using the equations in Ref. [1]. Assuming that photon detection probability is $\eta_d = 0.7$, the success probability is calculated to be 1.5×10^{-11} . On the other hand, using SFG-BSA, the success probability is calculated to be 1.2×10^{-11} with the SFG efficiency of 2.3×10^{-8} used in our experiment and assuming $\eta_d = 0.7$ as before, which is almost the same as that in linear optical implementation. Here, we assume that a complete Bell-state measurement is performed at the SFG-BSA. Moreover, in the region of lower detection efficiency, the SFG-BSA becomes more efficient. For example, when $\eta_d = 0.5$, SFG-BSA is 16 times more efficient than the linear optical scheme [10].

Another interesting example showing the qualitative differences between LO-BSA and SFG-based BSA can be seen when employed as an entanglement heralder for a loophole-free Bell test with probabilistic photon sources. In a Bell-test experiment, the correlation of the measurement outcomes is typically character-

	LO-BSA	SFG-BSA
Achievable CHSH value	2.34	$2\sqrt{2}$
Necessary detection efficiency	0.911	0.68

Table 1: Comparison between LO-BSA and SFG-BSA

ized by CHSH value S , with the Bell-CHSH inequality stipulating that $|S| < 2$ within the framework of local realism. The primary obstacle in loophole-free Bell tests is that optical loss in the transmission channel creates a detection loophole. In this context, LO-BSA has been identified as one of the plausible schemes for achieving such a heralding [13, 14]. However, according to Ref. [14], in conventional linear optical entanglement swapping, it was shown that the maximum CHSH value was $S = 2.34$, and the detection efficiency of $\eta_d = 0.911$ is at least necessary for each of Alice’s and Bob’s detectors to observe the violation of the Bell’s inequality. However, our simulation shows that, by using the SFG-BSA, the maximum CHSH value can be close to the Tsirelson’s bound [15] of $S = 2\sqrt{2}$. In addition, the detection efficiency of $\eta_d = 0.68$ is sufficient to observe the violation, which significantly mitigates the requirements for Alice’s and Bob’s detection systems. These results are summarized in Table.1 and clearly show the qualitative difference between LO-BSA and SFG-BSA.

We added these discussions to the Discussion section and the details of the theoretical simulation performed in (ii) in Supplementary Note 7. In addition, we revised the introduction section to clarify the qualitative difference of the SFG-BSA from LO-BSA.

- The manuscript is well-structured and technically precise. Some explanations in the main text are dense, particularly in describing the experimental setup and theoretical background. The highly technical nature of the work, particularly the nonlinear optics and entanglement swapping protocol, may be challenging for non-specialists. The main text is overloaded with an excessive number of acronyms and abbreviations. The discussion section lacks clarity on how the work compares quantitatively with other approaches and its realistic path toward practical use. In our opinion, the Authors demonstrated a nice and highly challenging piece of quantum technology, which may draw the attention of the community, even though the performance of the SFG-based Bell-state analyzer, quantum teleportation, and entanglement swapping in real-world quantum communication protocols is yet quite far from being applicable. Unfortunately, the manuscript is very hard to read, especially for readers outside the quantum optics field. Consequently, the paper does not yet meet the broad interest threshold of Nature Communications without significant revisions. If the authors improve clarity, broaden accessibility, and better justify the impact, it might be considered.

We sincerely acknowledge the comment regarding the potential difficulty for

non-specialists to follow certain parts of the manuscript. However, the included content is essential for ensuring the completeness and self-contained nature of the paper. Further simplification or omission would, in our view, hinder rather than help the reader's understanding. To address this concern, we have added references that provide useful background on key concepts and abbreviations in quantum communication and information processing [16, 17]. We believe these resources will be helpful particularly for early-stage researchers and students entering the field. In addition, we refrained from using some abbreviations such as BSM and DOF because they were not so frequently referred in the manuscript.

In the revised manuscript, the impact of the results was much strengthened by describing comparisons with existing approaches. Moreover, we added the extent to which the efficiency of SFG needs to be improved for practical applications, which clarifies the realistic path toward practical use. We believe that the revised manuscript would meet the reviewers' requirements.

Reviewer 3

We would like to thank the reviewer for the effort and time spent reading our manuscript.

Reviewer 4

We would like to thank the reviewer for the effort and time spent reading our manuscript and posing comments and questions about the details of the theory and experiment, which improve the clarity of our manuscript.

- The manuscript "Experimental entanglement swapping through single-photon $\chi(2)$ nonlinearity" presents the first entanglement swapping experiment based on the nonlinear Bell-state analyser (sum frequency generation) operating at the single-photon level. Compared to previous experiments, this work involves two individual photons in the SFG process and demonstrates feasibility by directly measuring the polarisation utilising superconducting nanowire single-photon detectors (SNSPDs). In the past, there were two main obstacles hindering the progress in this direction, namely the low efficiency of SFG at the single-photon level and the low efficiency of SNSPDs. To overcome these challenges, the authors constructed a home-made doped PPLN waveguide and used SNSPD with a high quantum efficiency. The presented experimental results clearly demonstrate faithful entanglement swapping between photons and are well-supported by theoretical calculations including losses. Such a demonstration of quantum protocols based on nonlinear single-photon interactions may open the way to all-optical quantum information processing and communication. I would recommend to publish the paper after authors address the following minor comments:

We thank the reviewer for recommending the publication. Below, we provide point-by-point answers to the reviewer’s comments and questions.

- - The current efficiency of SFG process is about 10^{-8} which is slightly higher but comparable to previous single-photon SFG experiments [Phys. Rev. Lett. 113, 173601, 2014]. Is it possible to improve this efficiency in PPLN waveguides by better design? Or are the new materials and devices such as nanophotonic cavities [<https://arxiv.org/pdf/2411.15437>] the only way to enhance the strength of nonlinear interaction ?

We think there are two possible ways to improve the nonlinear efficiency of waveguides: first, by increasing the crystal length, and second, by further optical confinement. For the crystal length, the nonlinear efficiency increases as the square of the crystal length, but it becomes even more difficult to ensure uniformity of the waveguide. Thus, It is not practical to improve nonlinear efficiency by lengthening the crystals any further. For the optical confinement, attempts to improve the nonlinear efficiency by reducing the core diameter are now being actively conducted around the world, specifically, experiments using PPLN thin films. For example, in Refs [18] and [19], the conversion efficiencies per cm^2 have been reported to be about 100 times higher than ours. The remaining challenge is the large insertion loss and the short crystal length. For example, in [18], the conversion efficiency $2837 \text{ \%}/\text{W}/\text{cm}^2$, crystal length 0.53 cm and insertion loss of 4.0 dB have been reported, which leads to overall conversion efficiency of $317 \text{ \%}/\text{W}$. This value is still lower than ours ($\sim 900 \text{ \%}/\text{W}$). Nonlinear devices with resonator structures have the potential to dramatically improve nonlinear efficiency, which we are very interested in. In particular, it is interesting to see how [3] can achieve high nonlinear efficiency ($4.4 \times 10^4 \text{ \%}/\text{W}$) and low insertion loss (30 \%) at the same time. We note that comparing overall teleportation rate in [20] with the same input average photon number, ours is still higher, because of the high-speed generation and efficient detection of entangled photons. Thus, combining our photon generation and detection system and such a nonlinear resonator would be interesting in the future.

- - In the section "Robustness of SFG-BSA against optical losses", two types of fake success events are mentioned, but the probability only for the second one is shown. It would be useful to estimate the first one as well.

According to the comment, we revised the section **Robustness of SFG-BSA against optical losses** in page 5 as "In the presence of high channel loss, the dominant error event is that one photon among the three photons transmitted through the channel is lost. This type of event can be categorized into two distinct types. First, one photon from EPS I is lost, and one photon from each of EPS I and EPS II successfully arrives at the BSA, which results in a coincidence detection. Second, one photon from EPS II is lost, and a coincidence detection is induced by two photons from EPS I. Although the first error event commonly occurs in the linear-optical BSA and SFG-BSA, the second error

event can be excluded using the SFG-BSA. It is because SFG occurs only when at least one photon arrives from each EPS, which results in an increase in the robustness of the SFG-BSA against optical loss. Let γ^2 be the respective photon pair generation probability of EPS I and EPS II, and let t be the transmittance of each channel. Then, the probabilities of the first and second error events are given by $p_{f1} \propto 2\gamma^6 t^2(1-t)$ and $p_{f2} \propto \gamma^6 t^2(1-t)$, respectively. Thus, 1/3 of the error events are excluded by using SFG-BSA. More detailed analysis is given in Supplementary Note 1 and 2.”.

- - It would be helpful to present the mode profiles of the PPLN waveguides involved in SFG process and add more information about spectral characteristics of the interacting photons.

According to the comment, we added the information about the spatial profiles and spectral characteristics of photons in modes a , b and c in Supplementary Note 6, and revised the first paragraph in page 6 as “The estimated spatial profiles of photons a , b and c are shown in Supplementary Note 6.” and the first paragraph in page 11 as “The spectral distributions of photons in modes a , b and c are shown in Supplementary Note 6.” In addition, we have added the detail of the fabrication of PPLN/W in Supplementary Note 5.

References

- [1] N. Sangouard *et al.*, Phys. Rev. Lett. **106**, 120403 (2011).
- [2] J. Lu, M. Li, C.-L. Zou, A. A. Sayem, and H. X. Tang, Optica **7**, 1654 (2020).
- [3] J. Akin, Y. Zhao, Y. Misra, A. K. M. N. Haque, and K. Fang, Light: Science & Applications **13**, 290 (2024).
- [4] C. Schuck, G. Huber, C. Kurtsiefer, and H. Weinfurter, Phys. Rev. Lett. **96**, 190501 (2006).
- [5] M. Barbieri, G. Vallone, P. Mataloni, and F. De Martini, Phys. Rev. A **75**, 042317 (2007).
- [6] W. P. Grice, Phys. Rev. A **84**, 042331 (2011).
- [7] F. Ewert and P. van Loock, Phys. Rev. Lett. **113**, 140403 (2014).
- [8] M. J. Bayerbach, S. E. D’Aurelio, P. van Loock, and S. Barz, Science Advances **9**, eadf4080 (2023).
- [9] S. E. D’Aurelio, M. J. Bayerbach, and S. Barz, npj Quantum Information **11**, 37 (2025).
- [10] C. Śliwa and K. Banaszek, Phys. Rev. A **67**, 030101 (2003).

- [11] C. Wagenknecht *et al.*, Nature Photonics **4**, 549 (2010).
- [12] S. Barz, G. Cronenberg, A. Zeilinger, and P. Walther, Nature Photonics **4**, 553 (2010).
- [13] M. Curty and T. Moroder, Phys. Rev. A **84**, 010304 (2011).
- [14] Y. Tsujimoto *et al.*, New Journal of Physics **22**, 023008 (2020).
- [15] B. S. Cirel'son, Letters in Mathematical Physics **4**, 93 (1980).
- [16] J. L. O'Brien, A. Furusawa, and J. Vučković, Nature Photonics **3**, 687 (2009).
- [17] S. Slussarenko and G. J. Pryde, Applied Physics Reviews **6**, 041303 (2019), https://pubs.aip.org/aip/apr/article-pdf/doi/10.1063/1.5115814/19739502/041303.1_online.pdf.
- [18] X. Wang *et al.*, npj Quantum Information **9**, 38 (2023).
- [19] J. Zhao *et al.*, Opt. Express **28**, 19669 (2020).
- [20] J. Akin, Y. Zhao, P. G. Kwiat, E. A. Goldschmidt, and K. Fang, arXiv:2411.15437 (2024).